# MaAts, an Alkylsulfatase, Contributes to Fungal Tolerances against UV-B Irradiation and Heat-Shock in *Metarhizium acridum*

**DOI:** 10.3390/jof8030270

**Published:** 2022-03-08

**Authors:** Lei Song, Xiaoning Xue, Shuqin Wang, Juan Li, Kai Jin, Yuxian Xia

**Affiliations:** 1Genetic Engineering Research Center, School of Life Sciences, Chongqing University, Chongqing 401331, China; 201826021018@cqu.edu.cn (L.S.); 201726021003@cqu.edu.cn (X.X.); 202026021000@cqu.edu.cn (S.W.); 201626022016@cqu.edu.cn (J.L.); 2Chongqing Engineering Research Center for Fungal Insecticide, Chongqing 401331, China; 3Key Laboratory of Gene Function and Regulation Technologies under Chongqing Municipal Education Commission, Chongqing 401331, China

**Keywords:** alkylsulfatase, stress tolerances, *Metarhizium acridum*, mycopesticides

## Abstract

Sulfatases are commonly divided into three classes: type I, type II, and type III sulfatases. The type III sulfatase, alkylsulfatase, could hydrolyze the primary alkyl sulfates, such as sodium dodecyl sulfate (SDS) and sodium octyl sulfate. Thus, it has the potential application of SDS biodegradation. However, the roles of alkylsulfatase in biological control fungus remain unclear. In this study, an alkylsulfatase gene *MaAts* was identified from *Metarhizium acridum*. The deletion strain (Δ*MaAts*) and the complemented strain (CP) were constructed to reveal their functions in *M. acridum*. The activity of alkylsulfatase in Δ*MaAts* was dramatically reduced compared to the wild-type (WT) strain. The loss of *MaAts* delayed conidial germination, conidiation, and significantly declined the fungal tolerances to UV-B irradiation and heat-shock, while the fungal conidial yield and virulence were unaffected in *M. acridum*. The transcription levels of stress resistance-related genes were significantly changed after *MaAts* inactivation. Furthermore, digital gene expression profiling showed that 512 differential expression genes (DEGs), including 177 up-regulated genes and 335 down-regulated genes in Δ*MaAts*, were identified. Of these DEGs, some genes were involved in melanin synthesis, cell wall integrity, and tolerances to various stresses. These results indicate that *MaAts* and the DEGs involved in fungal stress tolerances may be candidate genes to be adopted to improve the stress tolerances of mycopesticides.

## 1. Introduction

The use of biological pesticides to protect plants is becoming increasingly widespread in various agricultural crops [1]. Entomopathogenic fungi have shown a great potential to reduce the application of chemical pesticides due to their advantages, such as low possibility to induce insect resistance, environmental friendliness, and safety [1,2,3,4]. To date, about 80 companies worldwide have developed more than 170 kinds of pesticide products based on entomopathogenic fungi [5]. For instance, entomopathogenic fungi *Metarhizium* spp. have been successfully applied to control locust [6,7,8], fruit fly [9], and grasshoppers [10]. The available genome sequences have made the *Metarhizium* genera become one of the important model fungi for exploring some questions about insect fungal conidiation, stress tolerances, and pathogenesis [11].

Conidia are the asexual propagules in many entomopathogenic fungi and usually the infective unit of mycopesticides [12,13]. Thus, the conidial pathogenicity, conidial yield, as well as sensitivities of conidia to various adverse conditions are all critical for the production cost and the application efficiency of the mycopesticides [14,15]. Among the adverse factors, the high-temperature and the ultraviolet (UV) irradiation from sunlight can lead to the decrease of the conidial vitality and are closely related to the application efficiency of the mycopesticides [16,17,18,19]. The UV irradiation usually results in DNA mutation and damage, as well as protein denaturation, and it also damages a variety of intracellular substances [17,20,21]. In addition, CPDs, 6-4PPs, and its isomer transformed from DNA can cause cell mutation and even death by bending DNA double helix to make different angles [22,23,24]. High temperatures can cause protein denaturation and DNA damage through base loss, leading to depurination and membrane disorganization [25,26,27,28]. However, 20 to 30 °C is a suitable temperature for conidial germination, growth, and the conidiation of most entomopathogenic fungi [29]. Therefore, exploring the mechanisms of the conidial tolerances to UV irradiation and heat-shock will be helpful to improve the quality of conidia, which is important for the effectiveness and sustainability of the mycopesticides in pest control.

As a widespread substance in nature, sulfate esters can be used by many microorganisms as a sulfur source for growth by hydrolyzing the sulfate esters using sulfatases [30]. At present, three classes of sulfatases, including type I, type II, and type III sulfatases, have been identified [31]. Type I sulfatases can utilize one equivalent of water to cleave the RO–SO_3_^−^ in the process, which is the largest group among three classes of sulfatases [31]. Type I sulfatases contain a highly conserved sulfatase motif, C/S-X-P-X-A-XXXX-T-G [32], in the N-terminal region and a unique active-site aldehyde residue (the left C/S), α-formylglycine (FGly), which is installed post-translationally [33]. So far, fifteen of the type I sulfatases have been identified in humans [33]. The research found it related to the disease and the regulation of embryonic development [33,34]. Type II sulfatases release inorganic sulfates and the aldehyde by cleave sulfate esters and require α-ketoglutarate as a cosubstrate in this process [35,36,37]. Type III sulfatases contain a metallo-β-lactamases catalytic domain in the N-terminal, the C-terminal, and the central domain related to recruit substrates and the resistance to SDS [35]. The type III sulfatases usually activate a nucleophilic water molecule by a Zn^2+^ cofactor [31,38]. Mutations of residues Tyr246 and Gly263 of SdsAP, a type III sulfatase in *Pseudomonas* sp. S9, show that the mutants abolish the enzyme activity for SDS degradation, indicating that these residues are important for the functions of type III sulfatases [39].

To date, type III sulfatases were found and identified mainly in microorganisms, with a focus on the identification of the crystal structure [39] and the activities in degrading a surfactant [40]. In *Pseudomonas* sp. ATCC19151, alkylsulfatase could cleave alkyl-sulfate, such as SDS, and was named as SdsA accordingly [41]. In *Pseudomonas* sp. C12B, it could secrete up to five different alkylsulfatases, and these exert hydrolysis of many kinds of alkyl-sulfate substrate [42]. Alkylsulfatase from *P. aeruginosa* PAO1 has a wide substrate specificity; it could not only degrade long chain alkyl-sulfate but also short chain ones, such as decyl-sulfate, octyl-sulfate, and hexyl-sulfate [35]. The disruption of the alkylsulfatase gene causes the inactivation of that enzyme, rendering it unable to degrade SDS [35]. Beyond that, some bacteria isolated from wastewater showed the capacity to biodegrade alkylsulfatase [43]. In eukaryon, research related to alkylsulfatase is limited. Alkylsulfatases from *Saccharomyces cerevisiae* surprisingly degrade certain ary-sulfates, and this strain could utilize SDS regarded as a sulfur source to grow [44,45,46]. However, the functions of the type III sulfatases in filamentous fungi are still a mystery.

In this study, we characterized a type III sulfatase gene *MaAts* and presented the functional analysis of alkylsulfatase in *M. acridum*. We have found, unexpectedly, that the loss of *MaAts* delayed conidial germination and conidiation, but it had no effect on conidial yield and fungal virulence. In addition, the fungal tolerances to UV-B irradiation and heat-shock were significantly decreased after the deletion of *MaAts*. Furthermore, digital gene expression (DGE) profiling results showed the disruption of *MaAts* affected the expression levels of some genes related to melanin synthesis, cell wall integrity, and tolerances to various stresses.

## 2. Materials and Methods

### 2.1. Strains and Cultivation

*M. acridum* CQMa102 (wild type) has been deposited in the Genetic Engineering Center of Chongqing University. Nutrient-rich one-quarter SDAY medium was used for fungal cultivation. It contains 2.5 g peptone, 5 g yeast extract, 10 g dextrose, and 18 g agar per liter. Nutrient-poor Czapek-Dox medium was used to screen fungal transformants, and it contains 2 g NaNO_3_, 0.01 g FeSO_4_·7H_2_O, 0.5 g MgSO_4_·7H_2_O, 1 g KH_2_PO_4_, 1 g KCl, 30 g sucrose, and 15 g agar, per liter. *Escherichia coli* DH5α was prepared to propagate plasmid and grown on LB medium, which contained 5 g yeast extract, 10 g tryptone, 10 g NaCl, and 18 g agar per liter. *Agrobacterium tumefaciens* strain AGL-1 was adopted in fungal transformation [47].

### 2.2. Construction of MaAts Mutants

We used pK2-PB with phosphinothricin resistance gene and pK2-sur chlorimuron ethyl resistance gene to construct knockout and complementation strains, respectively [48]. For the deletion of *MaAts*, the upstream and downstream sequences of the *MaAts* gene were amplified with primer pairs MaAts-LF/MaAts-LR and MaAts-RF/MaAts-RR. The confirmed fragments were ligated to the pK2-PB vector to build pK2-PB-*MaAts*-LR (the targeted gene disruption vector). Then, the resulting vector was transformed into WT by *A. tumefaciens* mediated transformation. Nutrient-poor Czapek-Dox medium with phosphinothricin (500 μg/mL) was prepared to select fungal transformants. For the complementation of *MaAts*, the full length of *MaAts* (containing promoter 2.0-kb) was cloned with primer pair MaAts-HF/MaAts-HR. The fragment was ligated into pK2-sur vector to generate pK2-sur-*MaAts*. Then, it was transformed into Δ*MaAts* by *A. tumefaciens* mediated transformation. Nutrient-poor Czapek-Dox medium with chlorimuron ethyl (20 μg/mL) (Sigma-Aldrich, Bellefonte, PA, USA) was prepared to screen fungal transformants. All transformants were verified by PCR and Southern blotting. Primers used in this study are shown in Appendix A.

### 2.3. Southern Blotting

About 5 μg of genomic DNA from various fungal strains was digested with *Nru* I and *Xho* I, respectively. The digested fragments were separated, then transferred into a nylon membrane. The probe was prepared by cloning 500 bp fragment from genomic sequence of *MaAts* with primers MaAts-PF/MaAts-PR (Appendix A). DIG High Prime DNA Labeling, and Detection Starter Kit I (Roche, Mannheim, Germany) was used to label the probe.

### 2.4. Assessments of Conidial Germination and Conidiation Capacity

The conidial germination assays were conducted as described previously [49]. For the conidial yield assays, aliquots of 2 μL 10^7^ conidia suspensions of each strain were vertically spotted into 24-well plates supplied with 1 mL 1/4 SDAY medium per well, which were incubated at 28 °C for 3, 5, 7, 9, 11, 13, and 15 days, respectively. The conidia in each well were collected and then suspended in ddH_2_O containing 0.05% Tween-80. The total number of conidia in each well was counted by a haemocytometer. All experiments were repeated three times.

### 2.5. Assessments for Fungal Stress Tolerances

The assessments for fungal stress tolerances were performed as described previously [50]. For the fungal tolerances to UV-B irradiation, 50 μL conidial suspensions (1 × 10^7^ conidia/mL) of each strain were spread on nutrient-rich 1/4 SDAY plates, which were treated by 1350 mW/m^2^ UV-B irradiation for 1.5, 3.0, 4.5, and 6.0 h, respectively, then cultivated in darkness at 28 °C for 20 h. For the fungal tolerances to heat-shock, sterile 1.5-mL centrifuge tubes containing 100-μL aliquot of conidial suspensions (1 × 10^7^ conidia/mL) of different fungal strains were exposed in a water bath at 45 °C for 3.0, 6.0, 9.0, and 12.0 h. After treatment, 50 μL conidial suspensions of WT, Δ*MaAts,* and CP strains were spread onto 1/4 SDAY plates, then incubated at 28 °C for 20 h. The conidial germination rate of each strain was measured by microscopic examination and the time for 50% inhibition time (IT_50_) was determined.

### 2.6. Bioassays

The bioassays were conducted by topical inoculation against *Locusta migratoria manilensis* (5th-instar nymphs) according to the method described previously [51]. In brief, aliquots of 3 μL 1 × 10^7^ conidia/mL conidial suspension of WT, Δ*MaAts,* and CP strains, and 3 μL paraffin oil were dipped on the locust pronotums. After the treatments, the locusts were fed on fresh corn leaves daily and kept at 28 ± 2 °C with 75% relative humidity and a 16:8 h (light–dark) photoperiod. The survival was recorded every 12 h until all locusts died. The experiment was carried out with three replicates with 30 locusts in each group, and the experiment was repeated three times. The mean median lethal time (LT_50_) was estimated for the WT, Δ*MaAts,* and CP strains.

### 2.7. Quantitative Reverse Transcription (qRT) PCR

We spread 100 μL conidial suspensions (1 × 10^8^ conidia/mL) on nutrient-rich 1/4 SDAY medium and collected the fungal cultures after 3-d cultivation at 28 °C. The total RNA of fungal samples were extracted using RNA Kit (CoWin Biosciences, Beijing, China). PrimeScript^TM^ RT reagent kit (Takara, Dalian, China) was used for the reverse transcription to generate cDNA with oligo-dT primer. The SYBR-Green PCR Master Mix kit (Bio-Rad, Foster City, CA, USA) was applied for qRT-PCR in the iCycler system (Bio-Rad, Hercules, CA, USA). The *Magpd* (GenBank accession No. XM_007817733.1) gene was used as the internal standard. Using the 2^−ΔΔCT^ method, we assessed the transcript levels of target genes [52]. The experiment was replicated three times. All the primers for qRT-PCR are presented in Appendix A.

### 2.8. Alkylsulfatase Activity Assays

The conidia of each strain were harvested from nutrient-rich 1/4 SDAY medium after 3 d of growth at 28 °C and washed with ddH_2_O. The conidia were quick-frozen in liquid nitrogen and fully grinded (70 Hz for 3 min) using Tissue lyser-24 (Jingxin, Shanghai, China). Then, we transferred 100 mg conidial powder of each strain in a new 2 mL centrifuge tube. We added 1 mL of the extraction buffer (50 mM Tris-HCl buffer, 1 mM DTT, 1 mM EDTA, pH 7.5) and mixed well; we followed that with centrifugation (16,000× *g* for 10 min) to obtain the enzyme stock solution. The alkylsulfatase activities in different fungal strains were measured by the barium chloride-gelatin method [30]. In brief, 200 μL of enzyme stock solution and 200 μL substrate (sodium octyl sulfate) were mixed in reaction buffer (0.1 M Tris-HCl, pH 7.5). The final concentration of the substrate was 15 mM. The reaction was carried out at 30 °C for 10 min and stopped by adding trichloroacetic acid (50 μL of 15%, wt/vol). After centrifugation (800× *g* for 1 min), aliquots of 200 μL supernatant were added into the barium chloride-gelatin and mixed thoroughly. The blank control was 200 μL of the extraction buffer instead of 200 μL of the enzyme stock solution. The resultant mixtures were detected by measuring maximum absorbance at 360 nm. One unit of the alkylsulfatase activity was defined as ΔA360 = 0.001 after 10 min.

### 2.9. DGE Profiling

The raw data of DGE profiling based on biological triplicates have been deposited in the NCBI BioProject database (accession number: PRJNA753293). The RNA sequencing was conducted on the BGISEQ-500 platform by Beijing Genomics Institution (Wuhan, China). The differentially expressed genes (DEG) were identified with a fold change ≥ 2 and *q* value < 0.05. The DEG annotation was based on the NCBI protein databases and subjected to gene ontology (GO) analysis (http://www.geneontology.org/, accessed on 10 January 2022) for the enrichments of GO terms to three function classes (*p* < 0.05).

### 2.10. Data Analysis

All data obtained from the repeated experiments in each test were expressed as means ± standard deviations. The one-factor analysis of variance was adopted to analyze the experimental data based on three replicates. SPSS 17.0 (IBM, Armonk, NY, USA) was used to analyze the datasets in this study.

## 3. Results

### 3.1. MaAts Belongs to the Type III Sulfatase

In *M. acridum*, a type III sulfatase gene, *MaAts* (MAC_08440) was cloned. The *MaAts* comprised an ORF of 1890 bp encoding a protein consisting of 629-amino acid with predicted isoelectric point of 5.45 and molecular mass of 68.6 kD. This protein has no signal peptide and no transmembrane domain. The MaAts contains a β-lactamase domain at the N terminus, followed by an alkylsulfatase dimerization, and an alkylsulfatase C terminal region (Figure 1A). The phylogenetic analysis showed that *MaAts* belongs to the type III of sulfatase (Figure 1A). To examine the activity of the MaAts in *M. acridum*, we successfully generated the *MaAts*-disruption mutants as well as the complementation strains (Appendix A). The alkylsulfatase activity assays showed that the samples from the WT and CP strains exhibited significantly higher alkylsulfatase activities than those from Δ*MaAts* strain, and the alkylsulfatase activities were hardly detected in all the samples (WT, Δ*MaAts*, CP) with heat-shock treatment (Figure 1B).

### 3.2. Disruption of MaAts Delays Conidial Germination and Conidiation but Does Not Affect Fungal Virulence and Conidial Yield

To identify the biological function of *MaAts* on growth, we observed the colonies of each strain on nutrient-rich 1/4 SDAY medium by microscopy. Our results showed no significant difference between the fungal strains in growth rates (Appendix A). Interestingly, the conidial germination rate of Δ*MaAts* was significantly lower than the WT and CP strains (Figure 2A). The GT_50_ of the Δ*MaAts* (5.50 ± 0.06 h) strain was significantly longer compared to the WT (5.10 ± 0.02 h) and CP (4.98 ± 0.08 h) strains (*p* < 0.05; Figure 2B). In addition, we observed the whole conidiation process of each strain by microscopy. The results showed that the WT and CP strains had formed conidia at 16 h, while the Δ*MaAts* strain did not form conidia until 20 h (Figure 3A). At 34 h, the knockout strain produced a small amount of conidia, while the control strains (WT and CP) had produced numerous conidia. (Figure 3A). However, there was no significant difference in the conidial yield of each strain from 3 days to 15 days (Figure 3B).

To explore the virulence of fungal strains, we conducted bioassays by topical inoculation onto the pronotum of *L. migratoria manilensis* with conidial suspensions of each strain. As a result, there were insignificant differences in LT_50_ among each strain (*p* > 0.05), indicating that the *MaAts*-disruption did not affect the virulence of *M. acridum* (Appendix A).

### 3.3. Disruption of MaAts Reduced the Tolerances to UV-B Irradiation and Heat-Shock

To reveal the contributions of the *MaAts* gene to stress tolerances, the mature conidia (15-day-old conidia) of tolerances to UV-B irradiation and heat-shock were evaluated. The results showed that there is a dramatic difference in the conidial stress tolerances between the Δ*MaAts* and the control strains (Figure 4A). The inhibition time of 50% (IT_50_) of Δ*MaAts* (3.8 ± 0.09 h) was significantly reduced compared to control strains with the IT_50_s of 4.5 ± 0.1 h (WT) and 4.5 ± 0.2 h (CP), respectively (Figure 4B, *p <* 0.01). After the heat-treatment in time-course, the conidial germination rates of the control strains were significantly higher than those of Δ*MaAts* (Figure 4C, *p <* 0.01). The inhibition time of 50% (IT_50_) of Δ*MaAts* (8.1 ± 0.9 h) was significantly reduced compared to control strains with the IT_50_s of 11.7 ± 0.9 h (WT) and 10.1 ± 1.0 h (CP), respectively (Figure 4D, *p <* 0.01). The above results indicated that the deletion of the *MaAts* reduced conidial tolerances to UV-B irradiation and heat-shock.

Furthermore, the conidia with different maturity (3, 6, 9, 12, and 15-day-old) were respectively collected to determine their tolerances to UV-B irradiation and heat-shock. The results showed that the stress tolerances of all stages of the conidia from the Δ*MaAts* strain were severely compromised (Figure 4A,B, *p* < 0.01). Furthermore, the transcription levels of some genes related to fungal stress tolerances were detected by qRT-PCR in the 3-day-old conidia of the WT and Δ*MaAts* strains. The results showed that the transcription levels of seven genes were significantly reduced in Δ*MaAts* (Figure 5C).

### 3.4. Identification of DEGs Influenced by the MaAts

To further reveal the mechanism of the disruption of *MaAts* reducing the fungal tolerances to UV-B irradiation and heat-shock, the total RNA was isolated from the 3-day-old conidia of the Δ*MaAts* and WT strains to identify the DEGs between the Δ*MaAts* and WT strains via DGE profiling. As a result, 512 DEGs (fold change ≥ 2 and *p* ≤ 0.05) were identified from the DGE data. Of the DEGs, 335 DEGs were down-regulated and 177 DEGs were commonly up-regulated (Figure 6A). The details of DEGs are given in Appendix A. To further verify the reliability of the DGE data, we randomly selected 15 genes, including 5 up-regulated genes (MAC_06945, MAC_00193, MAC_04050, MAC_00228, MAC_03018) and 10 down-regulated genes (MAC_01777, MAC_07727, MAC_08480, MAC_07363, MAC_02161, MAC_01480, MAC_01513, MAC_06626, MAC_08198, MAC_06979), and detected their expression by qRT-PCR. For all the selected genes, the expression patterns obtained by qRT-PCR showed similar patterns of down-regulation and up-regulation, indicating that the DGE data were reliable (Appendix A). By gene ontology (GO) annotation, all DEGs (*p <* 0.05) were classified as cellular component, molecular function, and biological process, which were 160, 292, and 242, respectively (Figure 6B).

From the DGE data, dozens of DEGs involved in cell wall biosynthesis and stress tolerances were strongly affected by the disruption of *MaAts*. Seven DEGs related to plasma membrane and membrane fusion are down-regulated in Δ*MaAts*, including a rab GDP-dissociation inhibitor gene (MAC_00732), cytochrome P450 gene (MAC_01212), ABC multidrug transporter gene (MAC_01512), glucose-methanol-choline oxidoreductase gene (MAC_02934), VHS domain protein gene (MAC_03504), phosphate permease gene (MAC_05171), and tetraspanin gene (MAC_00860). Ten DEGs involved in cell wall biosynthesis were down-regulated, such as phospholipase A gene (MAC_06774), a putative maltase gene (MAC_00567), a putative oxidoreductase gene (MAC_01213), and so on (Appendix A). It is worth noting that two DEGs, the beta-glucosidase gene (MAC_00623) and a putative C2H2 finger domain protein gene (MAC_07727) related to fungal tolerances to heat were remarkably down-regulated in Δ*MaAts* (Appendix A). The PQQ-repeat-containing protein gene (MAC_07325), which could inhibit the melanin synthesis, was dramatically up-regulated in Δ*MaAts* (Appendix A). In addition, eight DEGs involved in oxidative stress were remarkably down-regulated in Δ*MaAts*, such as the phospholipase A-2-activating protein gene (MAC_00293), quinone oxidoreductase gene (MAC_02039), 6-phosphogluconate dehydrogenase, NAD-binding protein gene (MAC_01012), a putative ankyrin-repeat-containing protein gene (MAC_01774), and so on. Two DEGs, the guanyl-specific ribonuclease F1 gene (MAC_01777) and sugar transporter family protein gene (MAC_04267) related to salt stress were down-regulated in Δ*MaAts*. Furthermore, eleven DEGs related to other stresses in Δ*MaAts*, such as the cysteine synthase K/M:Cysteine synthase B gene (MAC_00603) and the transporter-like protein gene (MAC_04282) involved in aluminum tolerance in *Arabidopsis*. Thus, the absence of *MaAts* reduced the tolerances to UV-B irradiation and heat-shock, owing to the alteration of the transcription levels of genes involved in melanin synthesis, cell wall integrity, and tolerances to various stresses.

## 4. Discussion

Sulfatases are commonly divided into three classes: type I, type II, and type III sulfatases [31]. They can also hydrolyze the sulfate esters, which are widespread in nature, to generate a source of sulfur for growth in many microorganisms [30]. To date, the research of sulfatase largely focuses on type I sulfatase in bacteria and humans [33,40,53], while there is a little information about type III sulfatase. Ats, an alkylsulfatase, belongs to type III sulfatases and can hydrolyze the primary alkyl sulfate [31]. However, the roles of alkylsulfatase in a biological control fungus are still unclear. In this study, the roles of the alkylsulfatase gene *MaAts* were characterized in the model entomopathogenic fungus *M. acridum*. The results showed that the disruption of *MaAts* delayed the germination and the conidiation, while the conidial yield and fungal virulence were not affected. Interestingly, the conidial tolerances to UV-B irradiation and heat-shock were significantly impaired, which has not been reported before in other fungi. According to the DGE data, *MaAts* might affect the conidial tolerances to UV-B irradiation and heat-shock by influencing the transcriptional levels of some genes related to melanin synthesis, cell wall integrity, and tolerances to various stresses in *M. acridum*.

During the applications of mycopesticides in the field, many natural abiotic factors, especially UV irradiation and heat-shock, directly affect the effectiveness of fungal biocontrol agents [15,18]. In addition, the conidial tolerances to UV-B irradiation and high temperature are also served as the important indicators of the conidial quality during the industrial production of mycopesticides [54]. In this study, the results showed the loss of *MaAts* resulted in significantly reduced conidial tolerances to UV-B irradiation and heat-shock. Accordingly, the DGE data showed that the expression levels of some genes involved in fungal tolerances to different stresses were significantly changed in Δ*MaAts* compared to WT. The gene for C2H2 finger domain protein (MAC_07727), which was directly related to the response to cold, drought, and heat stresses in rice [55], was significantly down-regulated in Δ*MaAts*. The ROS levels in the fungal cell would rise after treatment with UV irradiation [16]. Among the DEGs, the ThiJ/PfpI family protein gene (MAC_08673), which was involved in the tolerance to ROS and specifically H_2_O_2_ and superoxide radicals in *Candida albicans* [56], were significantly down-regulated in Δ*MaAts*, suggesting that the Δ*MaAts* strain was more vulnerable to UV irradiation than the WT strain. Furthermore, a β-glucosidase gene (MAC_00623), which was involved in the thermotolerance of *Pyrococcus furiosus* [57], was significantly changed in Δ*MaAts*. Many reports have demonstrated that melanin synthesis and cell wall organization were both closely involved in fungal stress tolerances [58,59,60,61]. The PQQ-repeat-containing protein gene (MAC_07325), which could inhibit the melanin synthesis in *murine* [62], was dramatically up-regulated in Δ*MaAts*. In addition, ten cell wall-associated protein genes (MAC_06774 [63], MAC_00567 [64], MAC_01213 [65], MAC_01513 [66], MAC_02366 [67], MAC_02951 [68], MAC_04850 [69], MAC_06773 [70], MAC_00235 [71], and MAC_04881 [69]) were significantly down-regulated in Δ*MaAts*, which possibly resulted in a decrease in stress tolerances of *M. acridum*. The gene for feruloyl esterase B precursor (MAC_05632), which was involved in the cell wall composition and structure of plants [72], was significantly up-regulated in Δ*MaAts*. The vacuolar protein-sorting protein BRO1 gene (MAC_06773) was dramatically down-regulated in Δ*MaAts*, which was related to the regulation of secondary wall biosynthesis in *Arabidopsis* [70]. 

The loss of *MaAts* delayed germination and conidiation compared to the control strains. Consistently, some genes involved in germination, conidiation and fungal growth were down-regulated in Δ*MaAts* from the DGE data, such as the *GmcA* gene for the glucose-methanol-choline oxidoreductase (MAC_02934), which was involved in conidial germination and conidiation in *Aspergillus nidulans* [73]. Meanwhile, the fungal specific transcription factor gene *StuA* (MAC_06675) related to the growth of *A. nidulans* [74] was also down-regulated. In addition, the gene for the kelch repeat protein (MAC_09450), which could regulate cell apoptosis processes in *Drosophila* [75], was also down-regulated.

In summary, our results showed that the deletion of *MaAts* significantly decreased the conidial tolerances to UV-B irradiation and heat-shock. *MaAts* and some DEGs involved in fungal stress tolerances may be candidate genes to be adopted to improve the stress tolerances of mycopesticides. In the future, clarifying the roles of the DEGs identified in this study may be conducive to further reveal the molecular mechanisms of the fungal tolerances to UV-B irradiation and heat-shock.

## Figures and Tables

**Figure 1 jof-08-00270-f001:**
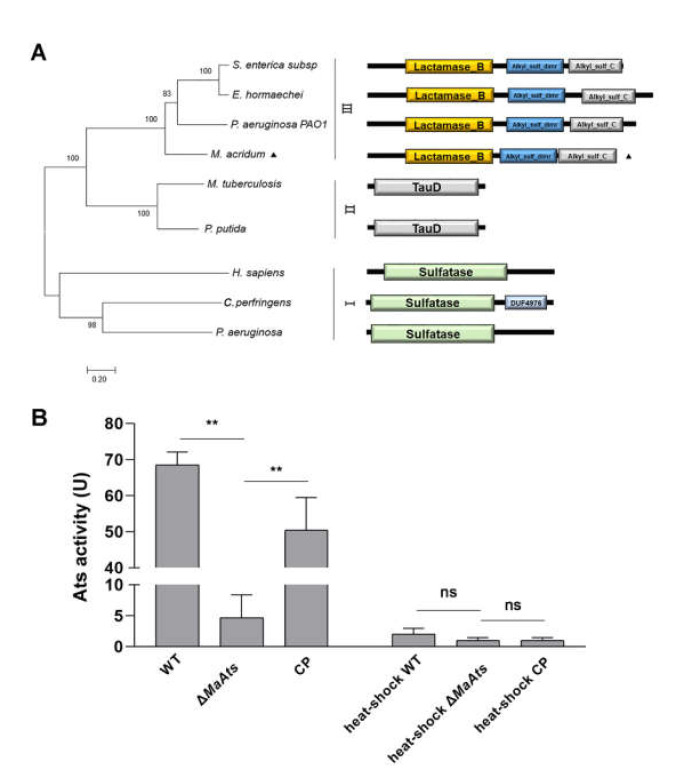
Features of MaAts in *M. acridum*. (**A**) Phylogenetic relationships of sulfatase proteins from bacteria, fungi, human. *S. enterica subsp*: *Salmonella enterica subsp* (EDZ09460.1); *E. hormaechei*: *Enterobacter hormaechei* (CBK85007.1); *P. aeruginosa* PAO1: *Pseudomonas aeruginosa* PAO1 (SKC13380.1); *M. acridum*: *Metarhizium acridum* (EFY85493.1); *M. tuberculosis*: *Mycobacterium tuberculosis* (NP_217923.1); *P. putida*: *Pseudomonas putida* (SUD73147.1); *H. sapiens*: *Homo sapiens* (NP_001012301.1); *C. perfringens*: *Clostridium perfringens* (WP_131337334.1); *P. aeruginosa*: *Pseudomonas aeruginosa* (SKC13380.1).I, II and III represent different types of sulfatases. (**B**) Alkylsulfatase activities in the 3-day-old conidia from nutrient-rich 1/4 SDAY medium. The heat-shock WT, heat-shock Δ*MaAts*, and heat-shock CP denotes that the samples were treated at 100 °C for 10 min. One unit of the alkylsulfatase activity was defined as ΔA360 = 0.001 after 10 min. Asterisk indicates significant difference at (**) *p <* 0.01, (ns) *p >* 0.05.

**Figure 2 jof-08-00270-f002:**
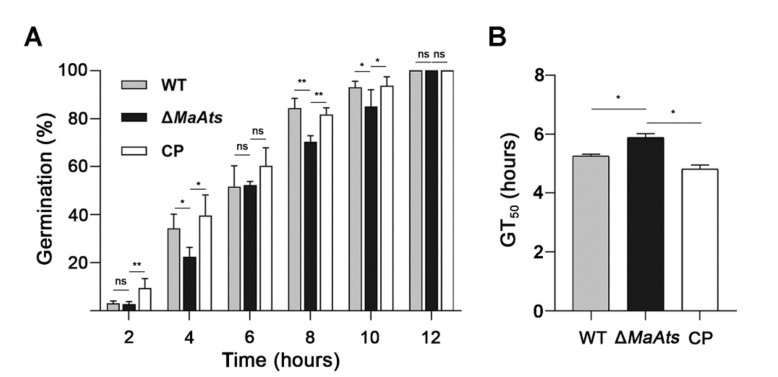
The deletion of *MaAts* delayed the conidial germination. (**A**) The germination rates of conidia of each strain on nutrient-rich 1/4 SDAY medium. (**B**) The mean 50% germination time (GT_50_) of each strain. Asterisk indicates significant difference at (*) *p <* 0.05, (**) *p <* 0.01, (ns) *p >* 0.05.

**Figure 3 jof-08-00270-f003:**
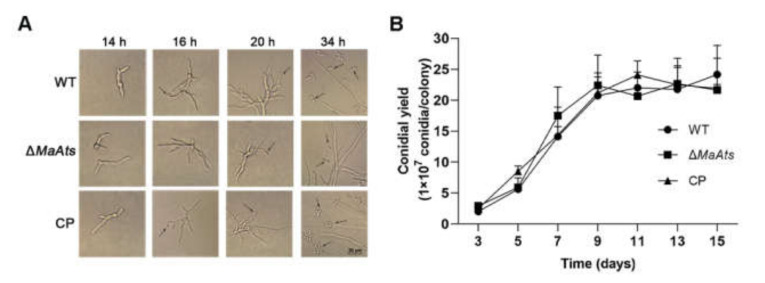
The deletion of *MaAts* delayed conidiation. (**A**) Conidiation of each strain cultured on nutrient-rich 1/4 SDAY medium. Black arrows indicate the typical conidiophores and conidia. (**B**) Conidial yield of WT, Δ*MaAts* and CP strains on nutrient-rich 1/4 SDAY medium at 28 °C.

**Figure 4 jof-08-00270-f004:**
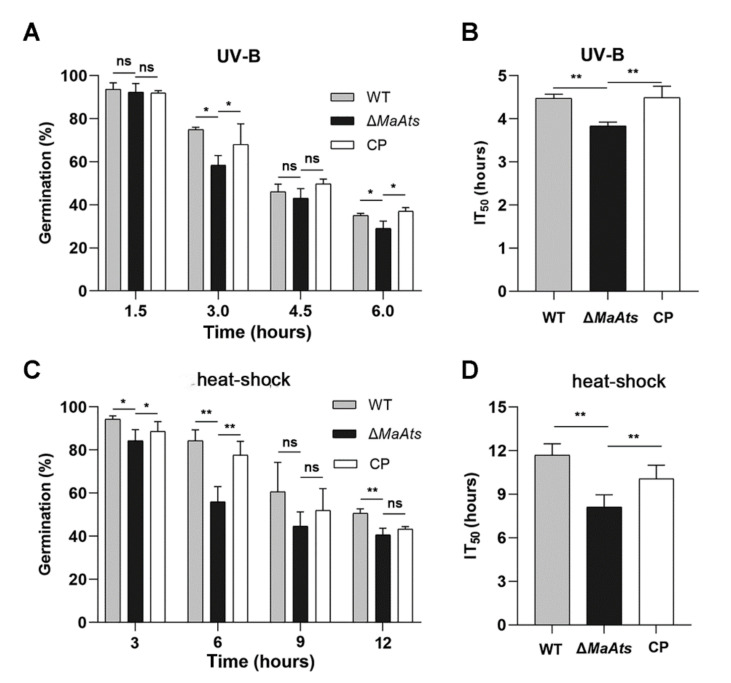
The deletion of *MaAts* impaired the UV-B irradiation and heat-shock tolerances of the mature (15-day-old) conidia. (**A**) Conidial germination after exposure of UV-B irradiation (1350 mW/m^2^) in a 1.5 h interval. (**B**) The half-inhibition time (IT_50_) under UV-B irradiation. (**C**) Germination rates after heat-shock treatment (45 °C) in a 3 h interval. (**D**) IT_50_ under high temperature (45 °C). After treatments, WT, Δ*MaAts* and CP strains were cultured on nutrient-rich 1/4 SDAY medium at 28 °C for 20 h. Asterisk indicates significant difference at (**) *p* < 0.01, (*) *p* < 0.05, (ns) *p* > 0.05.

**Figure 5 jof-08-00270-f005:**
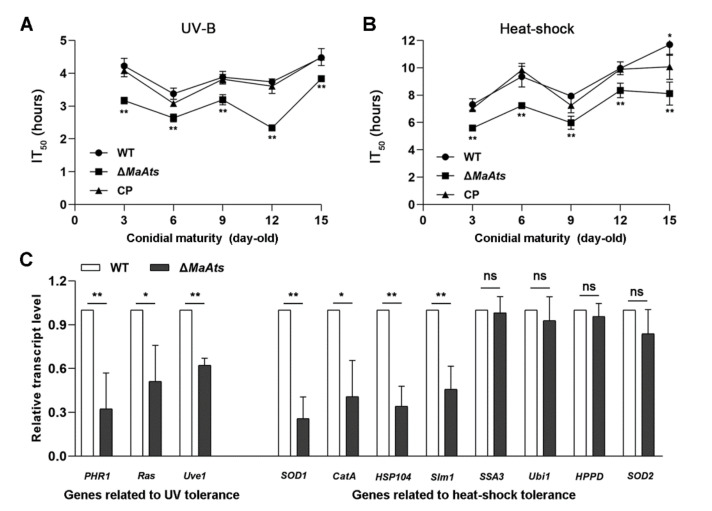
The deletion of *MaAts* diminished the UV-B irradiation and heat-shock tolerances of the conidia with different maturity. (**A**) The IT_50_ of conidia with different maturity (3, 6, 9, 12 and 15-day-old) under UV-B irradiation. (**B**) The IT_50_ of conidia with different maturity (3, 6, 9, 12 and 15-day-old) under heat-shock. (**C**) Relative transcript level of genes related to conidia tolerances to UV or heat-shock in the 3-day-old conidia. *MaPHR1* (MAC_05772); *MaRas* (MAC_07622); *MaUve1* (MAC_07337); *MaSOD1* (MAC_03040); *MaCatA* (MAC_09645); *MaHsp104* (MAC_05034); *Maslm1* (MAC_04714); *MaSSA3* (MAC_02927); *MaUbi1* (MAC_01946); *MaHPPD* (MAC_02015); *MaSOD2* (MAC_01660). Asterisk indicates significant difference at (**) *p <* 0.01, (*) *p <* 0.05, (ns) *p >* 0.05.

**Figure 6 jof-08-00270-f006:**
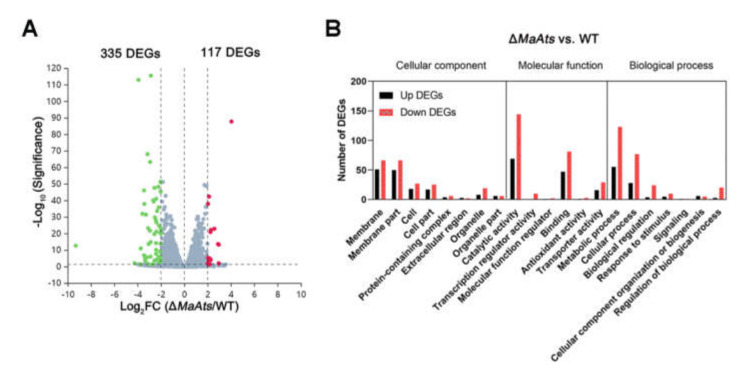
Identification and GO annotation of DEGs. (**A**) Identification of DEGs in the 3-day-old conidia from the Δ*MaAts* vs. WT. Green dots, down-regulated genes (log_2_ ratio ≤ 1). Red dots, up-regulated genes (log_2_ ratio ≥ 1). Grey dots, not differentially regulated (−1< log_2_ ratio < 1). (**B**) GO annotation of DEGs in the 3-day-old conidia from the Δ*MaAts* vs. WT.

## Data Availability

Not applicable.

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
