# Peer review of "MaAts, an Alkylsulfatase, Contributes to Fungal Tolerances against UV-B Irradiation and Heat-Shock in Metarhizium acridum"

_jof, 2022, doi:10.3390/jof8030270_

Round 1
Reviewer 1 Report
Dear Authors, your ms is very interesting but some points must be improved. M&M section must you add some things Results section must improve with add results which are missing. After you must fix the discussion. See me comments in the pdf.
Author Response
Thank you for your kindly suggestion. The point-by-point answers to the comments are as follows.
- Response: According to the comments, some mistakes have been corrected. (Lines 35, 37,135 in the revised manuscript)
- Response: In general, the mature conidia of can be obtained after Metarhizium grow on 1/4SDAY medium for 15 days. Thus, the conidial yields of different fungal strains were assayed every other day from the 3rd day to the 15th day after inoculation to fully reveal the role of MaAts gene in the conidial yield of Metarhizium acridum.
- Response: The details of locust rearing have been added in the revised manuscript as follows (Lines 160-162 in the revised manuscript):
After treatments, locusts were fed on fresh corn leaves daily and kept at 28 ± 2℃ with 75% relative humidity and a 16:8 h (light–dark) photoperiod.
- Response: The results of bioassays have been addressed in of the manuscript (lines 267-270 in the revised manuscript) and shown in supplementary materials (Fig. S2).
In addition, we tried our best to improve the manuscript and made some changes in the manuscript. These changes will not influence the content and framework of the paper. And here we did not list the changes but marked in blue in the revised manuscript.

Reviewer 2 Report
The manuscript "MaAts, an alkylsulfatase, contributes to fungal tolerances against UV-B irradiation and heat-shock in Metarhizium acridum" analyzes stress candidate genes in entomopathogenic fungi used in biological control of pest species. The need for the use of entomopathogenic fungi resides in the fact that there is a need to reduce the application of chemical pesticides and these organisms have interested characteristics such as low possibility of inducing resistance in insects, environmental compatibility, and safety.
In this study, it was demonstrated that MaAts and DEGs from M. acridum, involved in fungal stress tolerance may be candidate genes for improving mycopesticide stress tolerance. The authors after identifying a MaAts alkyl sulfatase gene constructed the deletion strain (ΔMaAts) and the complemented strain (CP) to reveal its functions in M. acridum.
Alkylsulfatase activity in ΔMaAts was drastically reduced compared with the wild-type (WT) strain. Loss of MaAts delayed conidia germination, conidiation, and significantly decreased fungal tolerances to UV-B irradiation and heat shock, whereas fungal conidial yield and virulence were unchanged in M. acridum.
Transcript levels of stress resistance-related genes were significantly changed after inactivation of MaAts. Furthermore, digital gene expression profiling showed that the differential expression genes identified, were both up-regulated (177) and down-regulated (355). Finally, the authors identified genes in ΔMaAts; some were involved in melanin synthesis, cell wall integrity, and ultimately tolerance to various stresses.
The work is certainly of high value, both for the results obtained and for the originality, high quality and completeness of the experimental design, clarity, and data representation.
The work has a high soundness for JoF and can be accepted in present form.
Author Response
Response: Thank you for your positive evaluation. We tried our best to improve the manuscript and made some changes in the manuscript. These changes will not influence the content and framework of the paper. And here we did not list the changes but marked in blue in the revised manuscript.

Round 2
Reviewer 1 Report
Dear authors thank you for the revised ms. I believe now it is ok